# Fecal Metagenomics Study Reveals That a Low-Fiber Diet Drives the Migration of Wild Asian Elephants in Xishuangbanna, China

**DOI:** 10.3390/ani13203193

**Published:** 2023-10-13

**Authors:** Xia Li, Junmin Chen, Chengbo Zhang, Shuyin Zhang, Qingzhong Shen, Bin Wang, Mingwei Bao, Bo Xu, Qian Wu, Nanyu Han, Zunxi Huang

**Affiliations:** 1Engineering Research Center of Sustainable Development and Utilization of Biomass Energy, Ministry of Education, School of Life Sciences, Yunnan Normal University, Kunming 650500, China; 2Southwest United Graduate School, Kunming 650092, China; 3Key Laboratory of Yunnan Provincial Education Department for Plateau Characteristic Food Enzymes, Yunnan Normal University, Kunming 650500, China; 4Xishuangbanna National Nature Reserve Management and Protection Bureau, Jinghong 666100, China; 5Asian Elephant Provenance Breeding and Rescue Center in Xishuangbanna, Jinghong 666100, China

**Keywords:** captive, wild, Asian elephant, fecal microbiota, diet, migration

## Abstract

**Simple Summary:**

The infrequent northward migration of wild Asian elephants in Xishuangbanna, China, has attracted global attention due to its rarity. Elephant migration is a complex ecological process, and the factors driving this long-distance migration remain the subject of ongoing research. In this study, by comparing the diets and fecal metagenomes of breastfed elephant calves, captive elephants, and their wild counterparts, we revealed that wild elephants exhibit a preference for crops and grains with a low fiber content. Furthermore, their gastrointestinal tracts harbor distinct microbial communities, functional hydrolases, and metabolic pathways tailored for processing these foods. These findings suggest that wild elephants have adapted to favor a low-fiber diet. This dietary preference, shaped by their unique gut microbiota, provides a significant explanation for the uncommon long-distance migration observed in wild Asian elephants.

**Abstract:**

The rare northward migration of wild Asian elephants in Xishuangbanna, China, has attracted global attention. Elephant migration is a complex ecological process, and the factors driving this long-distance migration remain elusive. In this study, fresh fecal samples were collected from both captive and wild Asian elephants, along with breastfed calves residing within the Wild Elephant Valley of Xishuangbanna. Our aim was to investigate the relationship between diet, gut microbiota, and migration patterns in Asian elephants through comprehensive metagenomic sequencing analyses. Among the breastfed Asian elephant group, Bacteroidales and *Escherichia* emerged as the dominant bacterial taxa, while the primary carbohydrate-active enzymes (CAZymes) enriched in this group were GH2, GH20, GH92, GH97, GH38, GH23, and GH43, aligning with their dietary source, namely breast milk. The bacterial taxa enriched in captive Asian elephants (CAEs) were mainly *Butyrivibrio*, *Treponema*, and *Fibrobacter*, and the enriched lignocellulose-degrading enzymes mainly included GH25, GH10, GH9, and cellulase (EC 3.2.1.4). These findings are consistent with the high-fiber diet of captive elephants. In contrast, the main bacterial taxa enriched in wild Asian elephants (WAEs) were *Ruminococcus* and *Eubacterium*, and the enriched CAZymes included GH109, GH20, GH33, GH28, GH106, and GH39. The abundance of lignocellulose-degrading bacteria and CAZyme content was low in WAEs, indicating challenges in processing high-fiber foods and explaining the low-fiber diet in this group. These findings suggest that wild elephant herds migrate in search of nutritionally suitable, low-fiber food sources.

## 1. Introduction

The Asian elephant (*Elephas maximus*) is a Class I protected wildlife species in China, is categorized as endangered on the International Union for Conservation of Nature (IUCN) Red List of Threatened Species™, and is encompassed in Appendix I of the Convention on International Trade in Endangered Species of Wild Fauna and Flora (CITES) [1,2]. China is home to approximately 300 wild Asian elephants (WAEs), concentrated solely in southern Yunnan Province, specifically within Xishuangbanna, Pu’er, and Lincang. Notably, 95% of the wild population resides within the Xishuangbanna National Nature Reserve [3]. Asian elephants require substantial space and access to abundant food and water sources. As the Asian elephant population in China has increased, so too have conflicts between humans and elephants. While previous studies have offered valuable insights into the dietary preferences [4], habitat requirements [5], and population dynamics [6] of Asian elephants, these conventional investigative approaches have played an important role in mitigating human–elephant conflicts. However, recent events involving a herd of WAEs have attracted global attention. These elephants traveled more than 500 km northward from the nature reserve to Kunming over a span of 3 months, traversing fields, roads, villages, and towns. No such long-distance migration of wild elephants has occurred in the past 50 years, and the underlying causes have not been determined. Elephant migration is a complex ecological process that varies in duration, distance, timing, and driving forces. Potential causes may include habitat shrinkage, food shortages, population growth, and the possibility of the lead elephant losing its way [3]. However, quantitative data to evaluate these factors are lacking. Therefore, studies on the causes of the migration of wild elephants are urgently needed.

The animal gut harbors large and complex microbial communities, resulting from long-term co-evolution between hosts and microbes [7,8]. These microbial communities participate in host immune regulation and nutrient uptake [8,9], playing important roles in the health and behavior of the host [10,11]. The study of the gut microbiota is a potentially important means to understand animal ecology, evolution, and conservation and is particularly important for conserving endangered species [12]. Several examples from animal studies illustrate the significant impact of microbes on host behavior. For example, the transfer of fecal microbiota from mice showing anxious behavior to germ-free mice resulted in timid behaviors in the hosts. When germ-free mice containing bacteria associated with anxiety were fed fecal pellets from control mice, they exhibited more exploratory behavior and similar behaviors to those of their fecal donor [13]. Gastrointestinal microbes manipulate host behavior under selective pressure to improve survival, sometimes at the expense of host fitness. *Toxoplasma gondii*, which suppresses the natural fear of cat odors in rats, provides a well-known example of how microbes influence host behavior in such a way that is detrimental to the host but beneficial to microbes that can invade new hosts [14]. Evidence shows that microbes can significantly influence behavior through the gut–brain axis [15]. Enteric nerves have receptors that respond to the presence of specific bacteria [16] and bacterial metabolites, such as short-chain fatty acids [17]. Therefore, it is worth exploring whether the gastrointestinal tract of WAEs contains specific bacteria and bacterial metabolites capable of stimulating enteric nerves, potentially triggering long-distance migration. In addition, individual members of the host gut microbiota and microbial communities have been shown to be highly dependent on the nutrient content of the host’s diet [18,19]. A consistent diet results in the enrichment of specific microbial taxa, potentially influencing host preferences for these foods [17]. Therefore, it is also necessary to investigate interactions between dietary nutrients, the gut microbiome, and behaviors of WAEs (foraging, migration, etc.).

In this study, we aimed to investigate the factors behind the migration of WAEs. We collected fresh feces of wild, captive, and breastfed elephants in the Xishuangbanna Nature Reserve for comparative metagenomic sequencing analyses. By combining the results of previous investigations on the plant components and nutrients of Asian elephant diets, we conducted a phylogenetic analysis of the fecal microbiota of Asian elephants, along with analyses of differentially abundant microorganisms and functions among groups. The findings were then correlated with the diet and migration of Asian elephants in an attempt to reveal the causes of the long-distance migration. The metagenomic studies of the gastrointestinal tract in this study provide a novel foundation for exploring the factors driving wildlife migration.

## 2. Materials and Methods

### 2.1. Sample Collection, DNA Extraction, and Sequencing

During the dry season in Xishuangbanna (November to April), the forest often experiences a shortage of food. However, there are natural water sources and an artificial salt station near the Wild Elephant Valley. This environment attracts wild elephants, leading them to frequently appear in the vicinity of the Wild Elephant Valley to forage. This facilitated the collection of fresh fecal samples for this study. In March 2019, fresh feces were collected from Asian elephants from the Wild Elephant Valley scenic area and the surrounding forest. Samples were collected from three calves that were breastfed in the captive group (BMST, BMSM, and BMTM), three completely captive adult elephants (CDD, CXG, and CFD) kept in captivity at fixed locations within the scenic area and mainly fed elephant grass, and three wild adult elephants (W2, W4, and W6). Wild and captive elephants had no opportunity for direct contact with each other. The collected samples were transported and preserved as detailed in previous reports [20,21]. DNA extraction and metagenomic sequencing were performed as described in our previous study [22].

### 2.2. Processing and Assembly of Raw Data

Fastp (https://github.com/OpenGene/fastp, version 0.20.0, accessed on 25 August 2023) was used to trim the adapter sequences at the 3′ and 5′ ends. Reads with a mean quality score below 20 and a length less than 50 bp, as well as N-containing reads, were removed to obtain high-quality paired-end and single-end reads. The reads were aligned against host DNA sequences using BWA (http://bio-bwa.sourceforge.net, version 0.7.9a, accessed on 25 August 2023), and reads with high similarity to these host sequences were removed. The optimized sequences were spliced and assembled using MEGAHIT [23]. Contigs ≥ 300 bp were screened as the final assembly results.

### 2.3. Gene Prediction and Abundance Analysis

Open reading frames within contigs were predicted using MetaGene [24] (http://metagene.cb.k.u-tokyo.ac.jp/, accessed on 25 August 2023). Genes with ≥100 bp were selected, and translated amino acid sequences were obtained. The predicted nucleic acid sequences of genes were clustered using CD-HIT [25] (http://www.bioinformatics.org/cd-hit/, version 4.6.1, accessed on 25 August 2023; parameters: 90% identity and coverage), and the longest genes were selected as representative sequences to construct non-redundant gene sets. SOAPaligner [26] (http://soap.genomics.org.cn/, version 2.21, accessed on 25 August 2023) was used to map the high-quality reads from each sample separately to the set of non-redundant genes (95% identity) to obtain the gene abundance in the corresponding samples.

### 2.4. Species and Functional Annotation

The protein sequences of the non-redundant gene set were aligned to the NR database (BLASTP alignment parameters: e-value of 10^−5^) using Diamond [27] (http://www.diamondsearch.org/index.php, version 0.8.35, accessed on 25 August 2023). The corresponding taxonomic database of the NR library was used to annotate the species. The amino acid sequences of the non-redundant gene set were aligned with the CAZy database (Carbohydrate-Active Enzymes Database) using hmmscan (http://hmmer.janelia.org/search/hmmscan, accessed on 25 August 2023), with the e-value threshold set to 10^−5^ to obtain functional annotations. Then, the sum of gene abundances corresponding to a carbohydrate-active enzyme (CAZyme) represented the abundance of the CAZyme. Protein sequences of the non-redundant gene set were aligned against the KEGG database (BLASTP alignment parameters: e-value of 10^−5^) using Diamond (http://www.diamondsearch.org/index.php, version 0.8.35, accessed on 25 August 2023) to obtain functional annotations. The sum of gene abundances corresponding to KEGG Orthology, modules, EC, and pathways was used to obtain the abundance of the corresponding functional category. All analyses were conducted using the online Majorbio Cloud Platform (https://cloud.majorbio.com/, accessed on 25 August 2023).

### 2.5. Statistical Analyses

Principal coordinate analysis (PCoA) was conducted for both the fecal microbiota and their functions. This analysis utilized the abundances of species and functional genes based on Bray–Curtis dissimilarity. PERMANOVA (adonis, permutations = 999) was used to assess intergroup differences in beta diversity. Statistically significant intergroup biomarkers were identified using linear discriminant analysis effect size (LEfSe). A strict all-against-all multiple comparison strategy was adopted, with an LDA score ≥ 4 and *p* < 0.05 indicating significance. The Kruskal–Wallis H test was performed to analyze differences in microbiota and CAZymes between groups, and *p* < 0.05 indicated significant differences. Tukey’s test was used to compare metabolic pathways and cellulases (EC 3.2.1.4) between groups, with *p* < 0.05 indicating significant enrichment.

## 3. Results

### 3.1. Diet Markedly Affects Asian Elephant Fecal Microbial Communities

A total of 1,044,465,766 raw reads (approximately 158 Gbp) were obtained via paired-end sequencing of the fecal metagenomes of Asian elephants. A total of 1,034,480,816 clean reads were obtained after quality control, and 1,034,412,974 optimized reads were obtained after de-hosting. Species annotation was performed by aligning the non-redundant gene sets against the NR database, and the results included five domains, eight kingdoms, 122 phyla, 271 classes, 625 orders, 1161 families, 3016 genera, and 14,060 species. Bacteria accounted for 99.45% in the breastfed Asian elephant (BAE) group, 98.81% in the captive Asian elephant (CAE) group, and 98.31% in the WAE group. These findings suggest that bacteria are the dominant microbes in Asian elephant gastrointestinal tracts [22].

Analysis of the fecal bacterial community species composition in Asian elephants at the phylum level (Figure 1a) showed distinct patterns. In CAEs, the dominant taxa were Firmicutes and Bacteroidota; in WAEs, the dominant taxa were Firmicutes, Bacteroidota, and Proteobacteria; and in BAEs, the dominant taxa were Bacteroidota, Proteobacteria, and Firmicutes. In the CAE group, the abundance of Spirochaetes (approximately 3.45%), Fibrobacteres (approximately 2.89%), and Synergistetes (approximately 1.04%) was higher than that of the WAE group. In the WAE group, the abundance of Actinomycetes (approximately 3.02%) and Verrucomicrobia (approximately 2.38%) was higher than that in the CAE group. The abundance of bacterial phyla other than the dominant phyla in the BAE group was consistently low.

The PCoA results at the genus level for the fecal microbiota of Asian elephants showed significant separation among the three sample groups (PERMANOVA: *R*^2^ = 0.93635, *p* = 0.005) (Figure 1b), indicating significant differences in the fecal microbial community structures of Asian elephants in the BAE, CAE, and WAE groups. A previous study, which examined the plant preferences and nutrient content of foods consumed by Asian elephants with different lifestyles [20], showed that CAEs mainly consume Napier grass (with neutral detergent fiber (NDF) content of approximately 72.8%). WAEs mainly forage for whole maize plants (with NDF content of approximately 49.5%), and the diet of calves in the BAE group consists only of breast milk. These dietary variations are likely crucial factors contributing to the significant differences in fecal microbial communities among the three groups of Asian elephants.

### 3.2. Enriched Microbial Taxa in Asian Elephant Feces Are Compatible with Their Diet

Based on the relative abundances of fecal bacteria in Asian elephants at different taxonomic levels within the BAE, CAE, and WAE groups, LEfSe was conducted to identify biomarkers with significant differences among the three groups (Figure 2a). Two bacterial phyla (Bacteroidetes and Proteobacteria) and one genus (*Escherichia*) were significantly enriched in the BAE group. Two phyla (Spirochaetes and Fibrobacteres) and three genera (*Butyrivibrio*, *Treponema*, and *Fibrobacter*) were significantly enriched in the CAE group. Two phyla (Firmicutes and Verrucomicrobia) and two genera (*Ruminococcus* and *Eubacterium*) were significantly enriched in the WAE group. A heatmap analysis of abundance clustering of differential species at the genus level (Figure 2b) showed that the differences in fecal microbiota in Asian elephants among the three groups were consistent with the findings of the biomarker analysis using LEfSe. These findings indicated that the significantly enriched microbial taxa in the feces of Asian elephants differed among groups.

In the BAE group, there was notable enrichment of Bacteroidales, particularly *Bacteroides fragilis*, a probiotic bacillus that colonizes the mammalian gut and is essential for maintaining host health [28]. The majority of *Escherichia* species are harmless and only cause intestinal infections under specific circumstances. Harmless *Escherichia coli* present in the intestines of calves fed on breast milk can ferment a variety of sugars, produce vitamins, and prevent the proliferation of pathogenic bacteria [29]. The microbial taxa with significant enrichment in the CAE group produced butyric acid using cellulose as a main substrate. Moreover, acetic acid, lactic acid, and other substances were also produced [30]. Some studies have reported that *Treponema* does not utilize cellulose. However, when co-cultured with *Fibrobacter succinogenes*, *Treponema* uses the soluble sugars released from cellulose by *F. succinogenes* as a fermentation substrate to enhance cellulose decomposition [31]. The interactions between *F. succinogenes* and *Treponema* promote the digestion of low-quality forage [32]. Therefore, the bacterial taxa with significant enrichment in the CAE group acted primarily to decompose cellulose and were compatible with the diet of the CAEs. *Ruminococcus* and *Eubacterium* were significantly enriched in the WAE group [33]. *Ruminococcus* are highly abundant in the human core gut microbiome and play a crucial role in the degradation and fermentation of dietary polysaccharides in ruminants [34]. For example, *Ruminococcus bromii* can specifically colonize the gut and degrade starch granules [35]. The enrichment of *Ruminococcus* in the intestinal tract of WAEs facilitated their digestion of maize and other crops grown on farmland. Therefore, the microbial taxa that were significantly enriched in the feces of BAEs, CAEs, and WAEs were compatible with the diets corresponding to these groups.

### 3.3. Enriched CAZymes Are Compatible with Asian Elephant Diet

A PCoA was performed to evaluate CAZymes in Asian elephants at the family level. The samples exhibited distinct clustering patterns corresponding to their dietary groups (PERMANOVA: *R*^2^ = 0.9107, *p* = 0.006) (Figure 3a). This indicated significant differences in the composition of CAZymes among BAEs, CAEs, and WAEs. To further investigate the effect of diet on glycoside hydrolases (GHs), a PCoA of GHs at the family level was performed. This analysis revealed even more pronounced clustering patterns based on different dietary groups, with highly significant differences (PERMANOVA: *R*^2^ = 0.91343, *p* = 0.004) (Figure 3b). Therefore, there were significant differences in the composition of GHs among the Asian elephant groups.

The results of the component analysis of CAZymes are summarized in Figure 4a. At the class level, GHs accounted for the highest proportions of 57.81%, 53.10%, and 53.96% in the BAE, CAE, and WAE groups, respectively. The abundance of glycosyl transferases, carbohydrate esterases, and carbohydrate-binding modules was above 9% in the BAE, CAE, and WAE groups. In the BAE group, the abundances of polysaccharide lyases (approximately 3.38%) and auxiliary activities (approximately 1.01%) were higher than those in the CAE and WAE groups. A compositional analysis of CAZymes at the family level is shown in Figure 4b. The GHs with the top five abundances in the BAE group were GH2, GH3, GH20, GH92, and GH31. The GHs with the top five abundances in the CAE group were GH2, GH3, GH109, GH31, and GH78. The GHs with the top five abundances in the WAE group were GH2, GH109, GH3, GH78, and GH20.

Based on the relative abundance of GH genes in the BAE, CAE, and WAE groups, intergroup differences in GHs at the family level were analyzed. Among the top 15 GHs with significant differences, GH2, GH20, GH92, GH97, GH38, GH23, and GH43 were significantly enriched in the BAE group. GH25, GH10, and GH9 were significantly enriched in the CAE group. GH109, GH20, GH33, GH28, GH106, and GH39 were significantly enriched in the WAE group (Figure 4c). The main substrate for GHs with significant enrichment in the BAE group was likely lactogenic glycoproteins containing mannose chains, as evidenced by the abundance of GH genes encoding mannosidase (Appendix A). The main functions of GHs that were significantly enriched in the CAE group were the degradation of xylan and dextran (Appendix A), suggesting that CAEs are adapted to a lignocellulose-rich plant-based diet. GHs with significant enrichment in the WAE group included polygalacturonase, rhamnosidase, xylosidase, arabinopyranosidase, glucanase, glucosidase, α-amylase, and arabinofuranosidase, but not cellulase and xylanase (Appendix A). This was related to the consumption of crops, grains, and melons with low fiber contents by wild elephants during extreme food scarcity.

### 3.4. Enriched KEGG Metabolic Pathways Are Correlated with the Dietary Nutritional Composition of Asian Elephants

Based on the Kyoto Encyclopedia of Genes and Genomes (KEGG) pathway database, hits were categorized into six groups with a total of 47 functional classifications (as shown in Appendix A). Among these, metabolism was the most abundant functional category in the metagenomic sequencing analysis of the intestinal tract of Asian elephants. Differentially expressed genes were involved in carbohydrate metabolism, enzyme families, amino acid metabolism, nucleotide metabolism, energy metabolism, and metabolism of cofactors and vitamins. The average relative abundances of metabolism, which contained 12 functions, were 69.39% in the BAE group, 66.04% in the CAE group, and 65.87% in the WAE group. Among the functions, carbohydrate metabolism accounted for 17.27%, 15.27%, and 15.87% in the BAE, CAE, and WAE groups, respectively. Enzyme families accounted for 10.89%, 10.97%, and 10.86%; amino acid metabolism accounted for 9.02%, 9.52%, and 9.09%; and nucleotide metabolism accounted for 5.81%, 7.15%, and 6.77% in the BAE, CAE, and WAE groups, respectively. These were the most abundant functional categories related to metabolism. Genetic information processing was the second most abundant functional category in the fecal metagenome of Asian elephants, with average relative abundances of 10.6%, 13.8%, and 13.68% in the BAE, CAE, and WAE groups, respectively. In particular, replication and repair exhibited average relative abundances of 5.63%, 7.34%, and 7.4% in the BAE, CAE, and WAE groups, respectively. Among the functional categories related to genetic information processing, translation accounted for 2.63%, 3.7%, and 3.66%, respectively. Environmental information processing was the third most abundant functional category in the intestinal metagenome of Asian elephants, with membrane transport accounting for 4.69%, 4.22%, and 4.29%, signal transduction accounting for 2.9%, 2.24%, and 2.19%, and signaling molecules and interaction accounting for 0%, 0.01%, and 0.01% in the BAE, CAE, and WAE groups, respectively.

Based on the KEGG metabolic pathway annotation results for the fecal metagenome analysis of Asian elephants, Tukey’s test was performed to analyze differences among groups (Figure 5). The results showed that galactose metabolism (ko00052), as well as fructose and mannose metabolism (ko00051), was significantly enriched (*p* < 0.05) in the BAE group. The substrates for the enzymes of these metabolic pathways were mainly lactogenic proteins with mannose as the sugar chain, indicating that the main food for calves may be lactogenic glycoproteins with mannose chains. The citrate cycle (ko00020) was significantly enriched only in the BAE group (*p* < 0.05), suggesting that lactogenic glycoproteins with mannose chains were sufficient for the energy requirements associated with the growth and development of calves. Leucine and isoleucine biosynthesis (ko00290) were enriched in the BAE and CAE groups, suggesting that the gut microbiota of young and captive elephants had a greater capacity to synthesize these essential amino acids than the gut microbes of wild elephants. Bacterial chemotaxis (ko02030) differed significantly among all three groups (*p* < 0.05) and was most abundant in the CAE group. Flagellar assembly (ko02040) was significantly enriched in the CAE group (*p* < 0.05). These findings suggest the susceptibility of CAEs to intestinal diseases. Flagellar motility and chemotaxis have been previously identified as important features of virulence in many pathogenic bacteria [36,37]. Genes involved in chemotaxis and virulence may work together to allow pathogenic bacteria to interact with specific tissues and, thus, invade the host [38], playing a very important role in pathogenesis. Compared to the BAE group, fatty acid degradation (ko00071) was significantly enriched in the CAE and WAE groups (*p* < 0.05), suggesting a lack of fatty acid catabolism in calves fed breast milk and in Asian elephants fed a plant-based diet. Metabolic pathways for cellulose degradation, including starch and sucrose metabolism (ko00500), were analyzed (Figure 5). This included key hydrolytic enzymes catalyzing cellulose to cellodextrins and then to cellobiose. Among these, significant intergroup differences were observed for cellulase (EC 3.2.1.4) (*p* < 0.05) (Figure 5), indicating that cellulase (EC 3.2.1.4) was the major cellulose hydrolase in the gastrointestinal tract of Asian elephants. Cellulase (EC 3.2.1.4) differed significantly between the CAE and WAE groups and was significantly enriched in the CAE group (*p* = 0.03703) (Figure 5). These findings suggest that captive elephants have a strong capacity to digest cellulose, while a low abundance of cellulose metabolic pathways in wild elephants can probably be explained by their consumption of crops and grains with a low fiber content.

## 4. Discussion

The causes of the long-distance northward migration of WAEs in the Xishuangbanna reserve have aroused the interest of researchers [3]. This study represents the first analysis of the migration of WAEs using a fecal metagenomics approach. Comparisons of the diet and fecal metagenome of wild elephants, breastfed calves, and captive elephants revealed that the interaction between diet and gut microbiota was an important cause of the migration of WAEs.

The survey of Asian elephant diets and the collected fresh fecal samples in this study were performed on the eve of their migration [20,21]. This study successfully acquired three exceptionally rare fresh fecal samples of BAEs. Additionally, on the same day, three distinct fresh fecal samples from WAEs were collected, ensuring that they were not mixed feces trampled by wild elephants. To maintain sample uniformity across groups, fresh fecal samples of three CAEs were collected. Although the sample size was small, the simultaneous acquisition of these scarce samples was an achievement. Since wild elephants are extremely aggressive and cannot be approached closely, the investigators were unable to distinguish individual elephants when collecting the fecal samples. Although this deficiency prevented us from performing further research on these three wild elephants, we could conduct corresponding research based on their populations.

The results of previous plant surveys showed that captive elephants mainly fed on Napier grass (*Pennisetum purpureum*), while wild elephants preferred low-fiber crops, grains, and melons [20]. An analysis of the nutrient contents of the diets of captive and wild elephants showed NDF contents of approximately 72.8% and 49.5%, respectively, suggesting that the diet of wild elephants was low in fiber. In addition, based on the statistical methods described by Ishwaran [39] for dietary analyses, the plant preferences of Asian elephants [40] included sugarcane (*Saccharum officinarum*), rice (*Oryza sativa*), maize (*Zea mays*), Napier grass (*P. purpureum*), *Thysanolaena latifolia*, sorghum (*Sorghum bicolor*), *Sorghum sudanense*, and *Musa acuminata* var. acuminata. However, the plant preferences of Asian elephants were unevenly distributed in small amounts, resulting in a very limited biomass. The lack of preferent plants in the wild leads wild elephants to forage for crops that are continuously planted in large areas. Asian elephants are particularly fond of foraging for palatable corn, rice, and sugarcane, which are concentrated and artificially cultivated at a large scale. Therefore, wild elephants forage for many local and cash crops that are artificially cultivated, and these crops are low in fiber. As a result, wild elephants prefer to forage for foods with a low fiber content.

The nutritional content of food contributes substantially to the gut microbiome of the host [41]. Consistent dietary patterns lead to the enrichment of specific microbial taxa and cause the host to develop a preference for these foods [17], eventually leading to compatible dietary habits. Bacteroidales, which was significantly enriched in the BAE group, is the major bacterial taxon in the gut of breastfed host infants and young children [42]. *Fibrobacter* and *Treponema*, which were significantly enriched in the CAE group, are mainly involved in cellulose degradation [31,32]. *Ruminococcus*, which was significantly enriched in the WAE group, is highly abundant in the core gut microbiota in humans [33] and plays a crucial role in the degradation and fermentation of dietary polysaccharides in ruminants [34]. For example, *R. bromii* could specifically colonize and degrade starch granules [35], suggesting that these specialized microbial communities contribute to the preference for grains and fruits in wild elephants. This suggests that the long-term interaction between dietary preferences and the gut microbiota has contributed to the development of a low-fiber diet in wild elephants. In addition, GHs enriched in the WAE group mainly included polygalacturonase, rhamnosidase, xylosidase, and α-amylase. In a KEGG enrichment analysis, cellulase (EC 3.2.1.4) was significantly less important in the WAE group than in the CAE group. These functional data further suggest that wild elephants in the Xishuangbanna reserve maintain a low-fiber diet over a long period and prefer cereal-based foods.

The low-fiber diet of wild elephants has led to a range expansion. This expansion has driven individual elephants to venture into human-inhabited areas in search of food, resulting in conflicts between humans and wild elephants [43,44]. In response to these challenges, efforts have been made to protect Asian elephants and increase awareness of wildlife conservation [45]. However, the number of wild elephants in China has recently risen significantly. The lack of palatable food for wild elephants in the Xishuangbanna reserve and the strict separation of areas for human and elephant activities have propelled the expansion of wild elephant habitats. In addition, Asian elephants are being hunted at a disturbing rate in Myanmar, south of the Xishuangbanna reserve [46]. Therefore, while ensuring the safety of the elephant herd, wild elephants, driven by their low-fiber diet, have migrated northward in search of preferred foods. They have stolen maize, rice, and other grains during migration to meet their dietary needs. Therefore, a diet low in fiber is likely the main reason for the long-distance migration of wild elephants from Xishuangbanna to the north.

## 5. Conclusions

In this study, a comparative analysis of the diets and fecal metagenomes of breastfed calves, captive elephants, and wild elephants was conducted. The results reveal that wild elephants prefer crops and grains with a low fiber content and that their gastrointestinal tracts are enriched with specific microbial taxa, functional hydrolytic enzymes, and metabolic pathways to digest these foods. These findings indicate that wild elephants have developed a preference for a diet with a low fiber content. Although this study did not evaluate the gut–brain axis or the mechanism by which the gastrointestinal microbiota regulates the migration of wild elephants via their diet, the preference for a low-fiber diet in wild elephants, driven by their unique gastrointestinal microbiota, offers a significant explanation for their long-distance migration. This study provides a new approach and basis for further gastrointestinal metagenomics analyses to determine the causes of wildlife migration. Furthermore, it provides theoretical insights for the conservation of WAEs and strategies to prevent abnormal large-scale migration.

## Figures and Tables

**Figure 1 animals-13-03193-f001:**
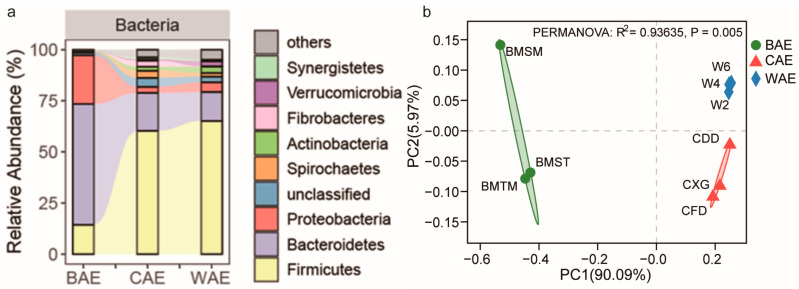
(**a**) The gut bacterial communities of Asian elephants at the phylum level. (**b**) PCoA of the gut microbial community of Asian elephants.

**Figure 2 animals-13-03193-f002:**
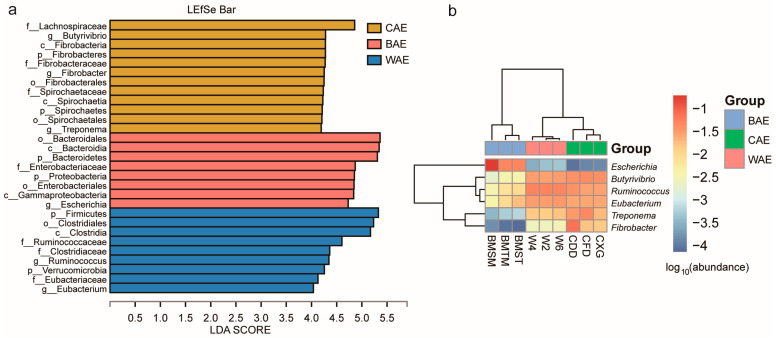
(**a**) LEfSe analysis of BAE, CAE, and WAE gut microbial communities. (**b**) Abundance heatmap analysis of BAE, CAE, and WAE gut taxa. The BAE group included three samples: BMSM, BMTM, and BMST. The CAE group included three samples: CFD, CXG, and CDD. The WAE group included three samples: W2, W4, and W6.

**Figure 3 animals-13-03193-f003:**
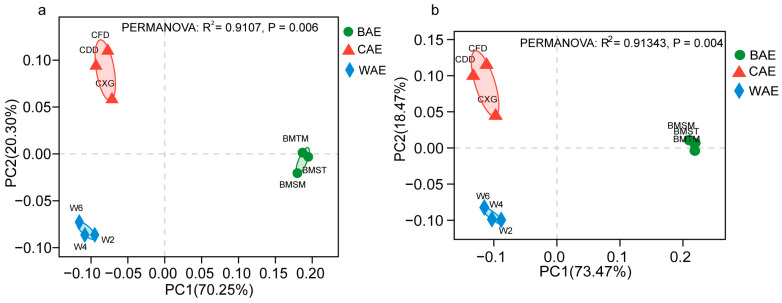
(**a**) PCoA of CAZymes in the BAE, CAE, and WAE groups. (**b**) PCoA of GHs in the BAE, CAE, and WAE groups.

**Figure 4 animals-13-03193-f004:**
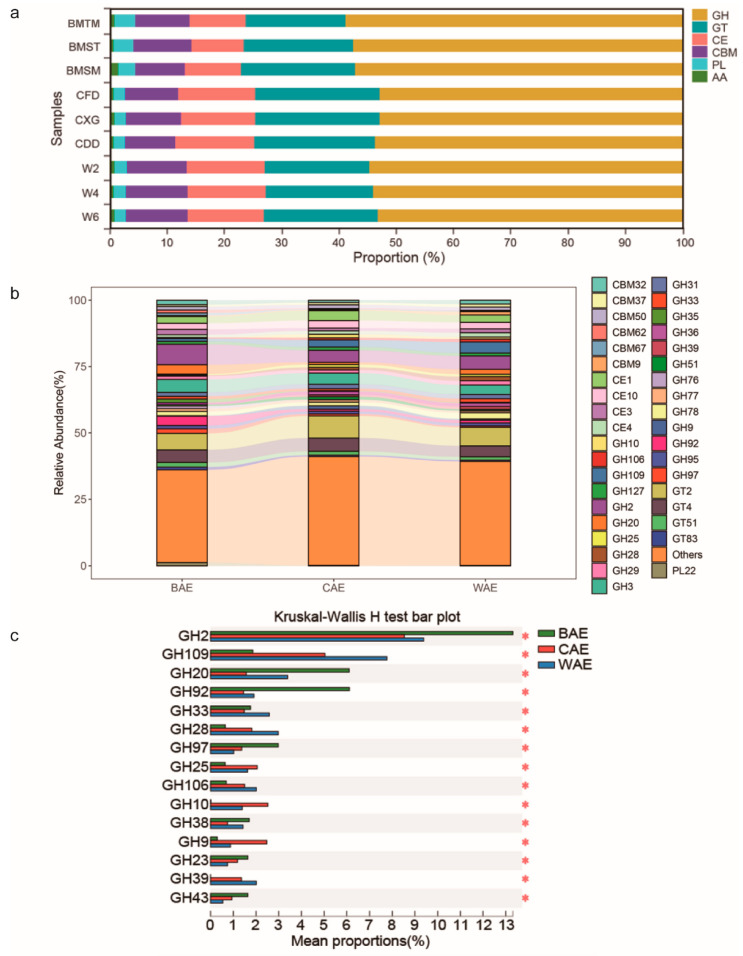
(**a**) Proportion of CAZymes at the class level. The BAE group included three samples: BMSM, BMTM, and BMST. The CAE group included three samples: CFD, CXG, and CDD. The WAE group included three samples: W2, W4, and W6. (**b**) Proportions of CAZymes at the family level. (**c**) Significant differences in GHs (family level) among BAEs, CAEs, and WAEs, * *p* < 0.05.

**Figure 5 animals-13-03193-f005:**
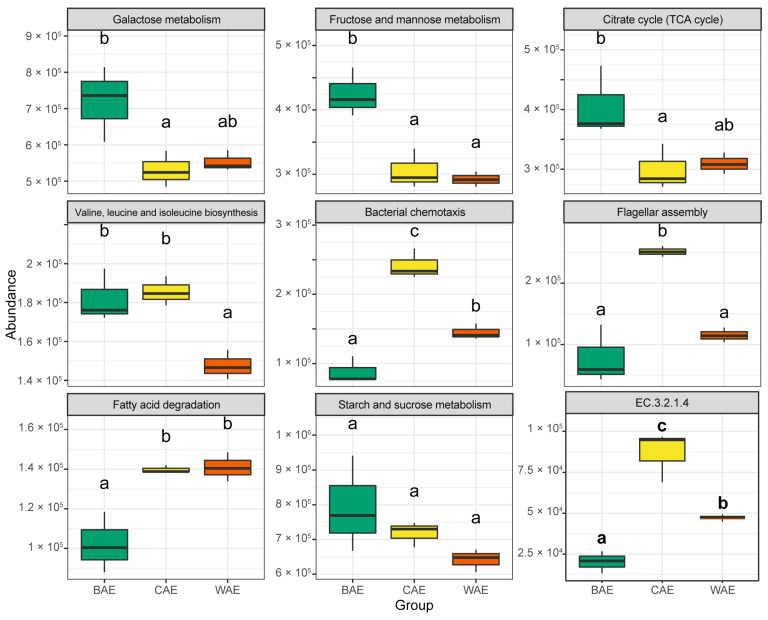
KEGG pathway enrichment analysis of Asian elephant fecal metagenome. a, b, c, and ab indicate the level of significant difference between groups, where the same letter indicates a lack of significant difference and different letters indicate a significant difference (*p* < 0.05).

## Data Availability

The raw sequence data obtained in this study have been deposited in the Genome Sequence Archive [47] at the National Genomics Data Center, Beijing Institute of Genomics, Chinese Academy of Sciences/China National Center for Bioinformation (GSA: CRA012186), and are publicly accessible at https://ngdc.cncb.ac.cn/gsa, accessed on 25 August 2023.

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
