# Peer review of "Fecal Metagenomics Study Reveals That a Low-Fiber Diet Drives the Migration of Wild Asian Elephants in Xishuangbanna, China"

_animals, 2023, doi:10.3390/ani13203193_

Round 1

Reviewer 1 Report

In this study, Li and colleagues provided an interesting explanation for elephant migration from the angle of gut microbiome by utilizing metagenomic analyses. However, I still have several concerns that need to be solved before this manuscript can be accepted.

1. While the authors have highlighted the significance of diet preferences in influencing the gut microbiome of elephants and subsequently their behavior, they have not presented any data (Figures or Tables) regarding the specific dietary choices of elephants.

2. The authors should tone down their claims: “Therefore, the preference for a low fiber diet caused by specific gastrointestinal microbiota provides a clear explanation for the long-distance migration of wild Asian elephants.”. I agree that diet preference may be one important reason for the migration of elephants, but it should not be the only one. If there is an ample supply of low-fiber food in the natural habitat of elephants, they may choose not to migrate.

3. Lines 137-138: This is not the right method to calculate species abundances. Please refer to “MetaPhlAn 4.0”.

4. Should be more specific about the index that you used to calculate gene abundances.

5. “flora” should be revised as “taxa”.

6. Line 190: No direct evidence was provided to support this claim; therefore the authors should revise “were” into “might be”, and remove “most”.

Author Response

Point 1: While the authors have highlighted the significance of diet preferences in influencing the gut microbiome of elephants and subsequently their behavior, they have not presented any data (Figures or Tables) regarding the specific dietary choices of elephants.

Response 1: Thank you for your comment. The main foraging plant species of completely captive and wild Asian elephants have been described in detail in our previous reports. Please refer to the reference (10.1016/j.gpb.2023.04.003), where Table S1 and Table S2 respectively display the diet of Asian elephants and the content of neutral detergent fiber (NDF) in their main food. Reference No. 20 (10.1016/j.gpb.2023.04.003) has been cited multiple times in this manuscript, lines 120, 370, and 383 ...

Point 2: The authors should tone down their claims: “Therefore, the preference for a low fiber diet caused by specific gastrointestinal microbiota provides a clear explanation for the long-distance migration of wild Asian elephants.”. I agree that diet preference may be one important reason for the migration of elephants, but it should not be the only one. If there is an ample supply of low-fiber food in the natural habitat of elephants, they may choose not to migrate.

Response 2: Thank you for your comment. We have used an significant explanation to replace a clear explanation in the lines 27 and 440 in the manuscript.

Point 3: Lines 137-138: This is not the right method to calculate species abundances. Please refer to “MetaPhlAn 4.0”.

Response 3: Thank you for your suggestion. We have removed the unreasonable expression Species abundance was estimated as the sum of gene abundances for the corresponding species. in line 144 in the revised manuscript. In addition, we have supplemented the online platform (Majorbio Cloud Platform) used for all analyses at lines 153-155 in the revised manuscript.

Point 4: Should be more specific about the index that you used to calculate gene abundances.

Response 4: Thank you, we used Reads Number_Relative to calculate gene abundances on the Majorbio Cloud Platform (https://cloud.majorbio.com/).

Point 5: “flora” should be revised as “taxa”.

Response 5: Thank you for your comment. we have revised “flora” to “taxa” in line 181 in the revised manuscript.

Point 6: Line 190: No direct evidence was provided to support this claim; therefore the authors should revise “were” into “might be”, and remove “most”.

Response 6: Thank you for your suggestion. We have revised this sentence in line 199 in the revised manuscript.

Reviewer 2 Report

This paper only discuss the effect of diet on gut microbiota, not migration. Many of parameters will change during the migration. I think this paper should focus on explaining the differences of gut microbiota between captive and wild elephant. The gut microbiota of adult species and calves is different, not only resulted from the diet. Unfortunately, a similar result has been published (10.1007/s10482-022-01757-1). The author should re-write the manuscript.

Metagenomic is not enough to exploring the bacteria at species level and also not enough to explaining the metabolic activity of microbes, which need to combine with metabolomic analysis. The author should add the metabolomic result.

Line 30. fecal sample, not lung samples.

Line 185-192. These sentences should put in the Discussion.

The classification level should consistent in the manuscript. Like Species composition at phylum level, PCoA at genus level, LefSe analysis at phyla, genus and species level, CAZymes at family level, etc. 

The author should improve the english language. 

Author Response

Point 1: This paper only discuss the effect of diet on gut microbiota, not migration. Many of parameters will change during the migration. I think this paper should focus on explaining the differences of gut microbiota between captive and wild elephant. The gut microbiota of adult species and calves is different, not only resulted from the diet. Unfortunately, a similar result has been published (10.1007/s10482-022-01757-1). The author should re-write the manuscript.

Response 1: Thank you for your comment. This study collected samples from wild Asian elephants on the eve of migration, rather than during the migration. We focused on studying the diet and fecal microbiota on the eve of migration, in an attempt to explore the factors that may lead to the migration of wild Asian elephants.

We strongly agree with your suggestion of many parameters will change during migration, but this study did not involve the research of wild Asian elephants during migration. Of course, the changes in fecal microbiome of Asian elephant during migration are a very attractive topic, and we will do our best to coordinate with relevant departments to try to obtain these samples for later research. 

This manuscript explored the fecal microbiome and diet of wild Asian elephants on the eve of their migration. We found that processing high fiber foods would be a challenge for individuals in the WAE group, which helps to explain their preference for choosing a low fiber diet. Therefore, we think that these data support the idea that the migration of wild Asian elephants was in search of preferred foods that are suitable for a low fiber diet.

In addition, the study on the differences in gut microbiota among different Asian elephant groups has been extensively explored in our previous papers (10.1016/j.gpb.2023.04.00310.3390/ani13050916, and 10.4014/jmb.1904.04033).

Point 2: Metagenomic is not enough to exploring the bacteria at species level and also not enough to explaining the metabolic activity of microbes, which need to combine with metabolomic analysis. The author should add the metabolomic result.

Response 2: Thank you for your suggestion. The metagenome assembly genomes (MAGs) at the strain level is already being processed, and the detection of the metaproteome and metabolome is also underway. The preliminary analysis results support our conclusions of this manuscript. The later analysis results will be considered for publication in this journal.

Point 3: Line 30. fecal sample, not dung samples.

Response 3: Thank you for your suggestion. We have revised dung samples” to “fecal samples in lines 31 and 113 in the revised manuscript.

Point 4: Line 185-192. These sentences should put in the Discussion.

Response 4: Thank you for your suggestion. This section is the description of the results of our previous research on the diet of Asian elephants, as well as the statement of the relationship between their diet and gut microbiota.

Point 5: The classification level should consistent in the manuscript. Like Species composition at phylum level, PCoA at genus level, LefSe analysis at phyla, genus and species level, CAZymes at family level, etc.

Response 5: Thank you for your comment. The display of species composition is usually at the phylum level (Figure 1a). The analysis of differences in species composition between groups is usually performed at a low taxonomic level, and this manuscript chose to focus on the genus level (eg. PCoA). Species classification (Domain, Kingdom, Phylum, Class, Order, Family, Genus, Species) and CAZymes hierarchy (Class and Family) are different and cannot be unified. LefSe analysis included all taxonomic levels of species (Figure 2a).

Round 2

Reviewer 2 Report

The species-level analysis should be deleted from this manuscript. 

Minor editing of English language required.

Author Response

Point1: The species-level analysis should be deleted from this manuscript.

Response:Thank you for your suggestion. The species-level analysis has been deleted from the revised manuscript. In particular, the species-level analysis in Figure 2 has also been removed. The results and discussion of the species-level analysis have been deleted or revised in lines 32, 36, 215, 217, 220, and 390. According to your suggestion, the quality of the revised manuscript has significantly improved.

Point2: Minor editing of English language required.

Response:We have thoroughly checked the English language of the entire manuscript, keeping in mind your comments. All revisions this time are marked in blue in the revised manuscript. According to your suggestion, the English language quality of the revised manuscript has significantly improved.